# Type F Quadricuspid Aortic Valve at Three-Dimensional Transoesophageal Echocardiography

**DOI:** 10.3390/diagnostics15212792

**Published:** 2025-11-04

**Authors:** Alessandro Barbarossa, Francesca Coraducci, Corrado Tagliati, Filippo Capestro, Barbara Orazi, Federico Guerra, Marco Di Eusanio, Antonio Dello Russo

**Affiliations:** 1Cardiology and Arrhythmology Clinic, Marche University Hospital, 60126 Ancona, Italy; alessandro.barbarossa@ospedaliriuniti.marche.it (A.B.); guerra.fede@gmail.com (F.G.); antonio.dellorusso@ospedaliriuniti.marche.it (A.D.R.); 2Cardiology Department, Carlo Urbani Hospital, 60035 Jesi, Italy; francescacorad@gmail.com; 3AST Ancona, Ospedale di Comunità Maria Montessori di Chiaravalle, Via Fratelli Rosselli 176, 60033 Chiaravalle, Italy; 4Cardiac Surgery Unit, Marche University Hospital, Marche Polytechnic University, 60126 Ancona, Italy; filippo.capestro@ospedaliriuniti.marche.it (F.C.); m.dieusanio@staff.univpm.it (M.D.E.); 5Territorial Cardiology, AST Macerata, 62100 Macerata, Italy; barbara.orazi@sanita.marche.it; 6 Department of Biomedical Sciences and Public Health, Marche Polytechnic University, 60126 Ancona, Italy

**Keywords:** quadricuspid aortic valve, type F, Hurwitz and Roberts, transoesophageal echocardiography, three-dimensional echocardiography, aortic regurgitation

## Abstract

We present the case of an asymptomatic 40-year-old male patient who underwent a transoesophageal echocardiography examination for further evaluation of aortic regurgitation. On physical examination, an occasional diastolic murmur was detected. Transthoracic echocardiography demonstrated aortic regurgitation that was difficult to quantify accurately. A type F quadricuspid aortic valve was subsequently identified and confirmed by three-dimensional transoesophageal echocardiography.

**Figure 1 diagnostics-15-02792-f001:**
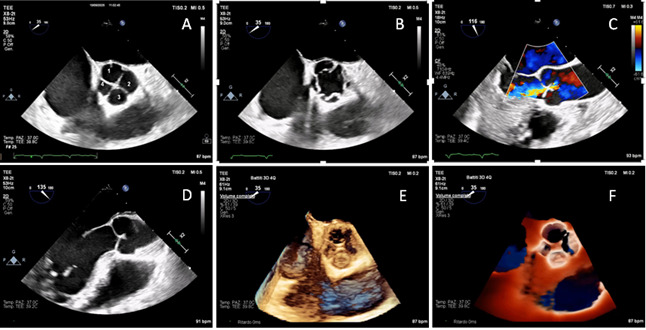
Transoesophageal examination was performed using a X8-2t (Philips, Amsterdam, The Netherlands) probe. Midesophageal aortic valve short-axis view (35° angle) demonstrated four cusps, with two equal-sized larger cusps and two unequal-sized smaller cusps, which means Hurwitz & Roberts type F quadricuspid valve (QAV) (**A**) [1,2]. Midesophageal aortic valve short-axis view in mid-systole showed the four commissures (**B**). Color Doppler midesophageal long-axis view demonstrated central moderate aortic regurgitation (vena contracta width = 4 mm, no holodiastolic flow reversal in proximal descending aorta) (**C**). Midesophageal long-axis view (135° angle) highlighted the characteristic leaflet coaptation pattern of QAV, with a coaptation length of 6 mm measured in end-diastole (**D**). Three-dimensional high resolution four-beats (61 Hz) standard reconstruction (**E**) and three-dimensional TrueVue (Philips, NL) rendering (**F**) provided a detailed visualization of cusps configuration supporting the diagnosis of QAV and its classification (Appendix A). QAV is a rare congenital abnormality with a prevalence of less than 0.05% in the general population [3,4]. Hurwitz and Roberts QAV classification is the most widely adopted one, and it identifies seven morphologic variants, from type A to type G [1,2]. This condition is frequently associated with progressive valve dysfunction, primarily due to isolated aortic regurgitation, while significant aortic stenosis is quite rare. The cumulative incidence of aortic valve intervention in QAV is comparable to that observed in bicuspid aortic valve. However, cumulative incidence of aortic surgery in patients with QAV is lower compared to those with bicuspid aortic valve, although it is higher than in the general population [2,5]. Aortic valve replacement with either a mechanical or a biological prosthesis can be performed. However, aortic valve repair should be considered as a first-line option, particularly in young patients with suitable anatomy, even though long-term durability data in this setting remain limited; tricuspidization of a QAV may be an option, particularly in cases with two larger and two smaller leaflets (Hurwitz and Roberts types F and G), or in the case of three larger and one smaller leaflets (Hurwitz and Roberts types B). In the literature, two cases of type F QAV associated with severe aortic regurgitation have been reported, without other significant valvular or aortic lesions. Both patients underwent aortic valve replacement [6,7]. If aortic dilatation is detected, Bentall or Ross procedures should be considered [8,9]. Anatomical studies of normal tricuspid aortic valves reported a mean coaptation length of 3.8 ± 0.8 mm [10]; however, corresponding data on QAVs are currently lacking. QAVs may show a greater coaptation length, potentially related to valve morphology, and future studies on this topic are needed.

## Data Availability

The original contributions presented in this study are included in the article/Appendix A. Further inquiries can be directed to the corresponding author.

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
