# Peer review of "Type F Quadricuspid Aortic Valve at Three-Dimensional Transoesophageal Echocardiography"

_diagnostics, 2025, doi:10.3390/diagnostics15212792_

Round 1
Reviewer 1 Report
Comments and Suggestions for Authors
I would like to thank you for your really interesting case and wonderful images and for your excellent language.
However, I have only one comment. In your "Institutional Review Board Statement", I think it is not enough to say "Not applicable". I think it would be better to write the reason why your work did not require instructional review board approval at to mention if your work was conducted in accordance with the Declaration of Helsinki.
Otherwise, your work is excellent and I have no other comment.
Author Response
Thank you very much for your comments
In general, single-patient case reports do not require Institutional Review Board approval, as they are typically not classified as research under regulations in the U.S. and Europe. The work was conducted in accordance with the Declaration of Helsinki.
Reviewer 2 Report
Comments and Suggestions for Authors
This manuscript presents a rare case of Type F quadricuspid aortic valve (QAV) with excellent image quality. The authors clearly demonstrate this rare anatomical structure and the associated moderate aortic regurgitation using 2D and 3D TEE techniques. Although the diagnosis of QAV using TEE has been well reported, the image quality in this case makes it particularly valuable for education. However, the manuscript has some issues regarding the completeness of the clinical description and the rigor of the echocardiographic quantitative analysis, which should be addressed before publication.
Major Comments
1. The manuscript diagnoses "moderate AR" based solely on the color Doppler image (Fig. 1C), which lacks the comprehensive quantitative assessment recommended by guidelines (such as ASE/ESC). The authors should provide 1-2 additional objective quantitative parameters (e.g., Vena Contracta width, PISA/EROA, regurgitant volume, diastolic flow reversal in the descending aorta, etc.) to support this diagnosis, or clearly state in the manuscript why these data could not be obtained.
Minor Comments
1. The abstract mentions the patient was "asymptomatic", but the purpose of the TEE was for "aortic regurgitation assessment". There is a slight contradiction between the two. Please add a brief clinical context (e.g., was the patient referred due to a heart murmur found on physical examination?) to make the clinical scenario more complete.
2. Spelling inconsistency: 1) "transoesophageal" and "transesophageal" are used interchangeably in the text; please unify them according to the journal's style. 2) "Robert" in the figure caption should be "Roberts".
3. The figure caption phrasing needs to be more rigorous: Figure 1D shows leaflet coaptation (and measurement of coaptation length); its cardiac phase should be specified as "diastole," just as Figure 1B is specified as "mid-systole".
4. The keywords "3D-echocardiography" and "three-dimensional" are semantically redundant; it is recommended to delete one of them.
Author Response
Thank you very much for your comments
Major Comment 1: We added the vena contracta and diastolic flow reversal for quantification
Minor Comment 1: Clinical scenario and transthoracic echocardiography were added
Minor Comment 2: Spelling inconsistencies were corrected
Minor Comment 3: diastole was added
Minor Comment 4: one semantically redundant word (3D) was deleted in keywords
Reviewer 3 Report
Comments and Suggestions for Authors
In the current image paper, the authors presented a patient with a type F QAV. The paper has no novel or interesting finding. There is no detailed discussion on differential diagnosis (e.g., distinguishing QAV from tricuspid valve with a raphe or commissural fusion variants). There is also no any interesting tips or tricks for therapetic options in the current case. Please include acquisition details (TEE probe type, imaging plane angles, frame rate, gain settings). The discussion would benefit from a brief comparison with previously reported Type F cases, particularly regarding associated lesions, regurgitation severity, and surgical management when indicated.
Author Response
Thank you very much for your comments
Probe type, details about acquisition planes and modality (multi-beat with high spatial resolution) were added.
Two references were added
Round 2
Reviewer 3 Report
Comments and Suggestions for Authors
The authors reasonably replied to all my previous comments. I have no further comments.